# In Vitro Hyperthermia Evaluation of Electrospun Polymer Composite Fibers Loaded with Reduced Graphene Oxide

**DOI:** 10.3390/polym12112663

**Published:** 2020-11-11

**Authors:** Ignacio A. Zárate, Héctor Aguilar-Bolados, Mehrdad Yazdani-Pedram, Guadalupe del C. Pizarro, Andrónico Neira-Carrillo

**Affiliations:** 1Departamento de Ciencias Biológicas, Facultad de Ciencias Veterinarias y Pecuarias, Universidad de Chile, Av. Sta. Rosa, La Pintana, 11735 Santiago, Chile; zarate.utem@gmail.com; 2Departamento de Química Orgánica y Físico Química, Facultad de Ciencias Químicas y Farmacéuticas, Universidad de Chile, Olivos, Independencia, 1007 Santiago, Chile; haguilar@ciq.uchile.cl (H.A.-B.); myazdani@ciq.uchile.cl (M.Y.-P.); 3Departamento de Química, Facultad de Ciencias Naturales, Matemática y del Medio Ambiente, Universidad Tecnológica Metropolitana, Av. José Pedro Alessandri, Ñuñoa, 1242 Santiago, Chile; gpizarro@utem.cl

**Keywords:** electrospun fibers, hyperthermia, electrospinning, polymers, near-infrared radiation (NIR), thermally reduced graphene oxide (TrGO)

## Abstract

Electrospun meshes (EM) composed of natural and synthetic polymers with randomly or aligned fibers orientations containing 0.5% or 1% of thermally reduced graphene oxide (TrGO) were prepared by electrospinning (ES), and their hyperthermia properties were evaluated. EM loaded with and without TrGO were irradiated using near infrared radiation (NIR) at 808 nm by varying the distance and electric potential recorded at 30 s. Morphological, spectroscopic, and thermal aspects of EM samples were analyzed by using SEM-EDS, Raman and X-ray photoelectron (XPS) spectroscopies, X-ray diffraction (XRD), and NIR radiation response. We found that the composite EM made of polyvinyl alcohol (PVA), natural rubber (NR), and arabic gum (AG) containing TrGO showed improved hyperthermia properties compared to EM without TrGO, reaching an average temperature range of 42–52 °C. We also found that the distribution of TrGO in the EM depends on the orientation of the fibers. These results allow infering that EM loaded with TrGO as a NIR-active thermal inducer could be an excellent candidate for hyperthermia applications in photothermal therapy.

## 1. Introduction

A composite material consists of different phases being homogeneous on a macroscopic scale and heterogeneous on a microscopic scale. These materials are basically made up of a matrix with a major proportion, which may be metallic, polymeric, or ceramic, and a reinforcement with minor proportion, which given its geometry, can be particles, fibers, or sheets.

Among reinforcing materials, carbon-based materials such as carbon nanotubes (1D) and graphene (2D) are the most used [1]. Given the structural and hybridization characteristics of graphene and its strong optical absorbance in near infrared region [2,3,4], it has been used as a photothermal agent for the in vitro and in vivo therapies in cancer treatment [5,6,7]. Different materials absorbing near infrared radiation (NIR) light to generate heat have been developed such as metallic and magnetic nanoparticles, conductive polymers, and carbon nanomaterials [8]. Graphene as a carbon nanomaterial can be subjected to oxidative and thermal processes, obtaining functionalized graphene as graphene oxide (GO) and thermally reduced graphene oxide (TrGO), respectively [9,10]. 

Photothermal therapy based on NIR (700–900 nm) has become a biomedical tool and an alternative to classical cancer therapies [11,12]. This technique allows the vibrational excitation through light absorption combined with laser-induced hyperthermia. Hyperthermia as cancer therapy, through heating, is used independently and in combination with chemotherapy for enhancing therapeutic efficacy. NIR light-mediated photothermal therapy, which has advantages such as a great capacity of heat ablation and less side effects, has emerged as a promising approach for cancer treatment. NIR light minimally interacts with physiological constituents such as proteins, coenzymes, or water and therefore negligently affects front wave propagation [13]. 

Composite materials can be made up of a mixture of natural or synthesized polymers, each one of which imparts particular characteristics. Thus, these materials could be composed of arabic gum (AG), which provides emulsifying stability and viscosity [14], natural rubber (NR) with excellent performance properties such as high elasticity and good fatigue resistance [15], and a synthetic polymer such as polyvinyl alcohol (PVA) that has good process ability and is easy to electrospun [16] through an electrospinning (ES) process. ES is a simple, low cost, and versatile fiber-spinning nanotechnology to generate electrospun polymer fibers [17]. These composite materials have attracted huge attention in various disciplines, particularly in biomedicine, because biomedical electrospun polymer fibers can be produced from either natural or synthetic polymers [18,19,20]. Biomedical EMs based on chemophotothermal combination therapy for improving cancer treatment is highly demanded due to their high efficiency for drug delivery and photothermal applications, as is the case of GO [8,20]. ES allows fibers to be obtained by means of coaxial stretching of a viscose polymer solution by applying an electric field [17]. All these compounds can be considered as biomaterials; that is, those natural or artificial materials that are implanted into a living organism in order to morphologically and functionally restore tissues or organs altered by trauma, malformations, or degenerative diseases [21]. EMs based on biopolymers have been recognized as a therapeutic system with great potential for ablation tumor [22], to prevent tumor recureence in vivo [23], cancer research [24,25], antibacterial properties [26], etc.

It is well known that new substrates, as electrospun polymer fibers produced from either natural or synthetic polymers by ES containing TrGO as NIR radiation to heat conversion enhancer in hyperthermia, are highly demanded in various biomedical disciplines [27]. To the best of our knowledge, polymer composite fibers with random or aligned fiber orientations in the EMs were prepared for the best time by using PVA, NR, AG stabilized by sodium dodecyl sulfate (SDS) as surfactant, and TrGO. The hyperthermia capacities of EMs were evaluated and compared with composites fibers without TrGO. Moreover, EMs were characterized by Raman, XPS, XRD, and SEM-EDS allowed to confirm the synthesis of TrGO, the composition of the polymer composite fibers, and the presence of TrGO on the fiber’s surface. In this paper, we will focus on the hyperthermic effect of 0.5% and 1.0% TrGO of EMs upon NIR irradiation recording by using electric potentials from 5 to 10 V and at a distance of 2 to 6 cm.

## 2. Materials and Methods

### 2.1. Material and Equipment

Reagents of the highest available grade were used. Ultra-pure water (18.2 MΩ) was obtained from the LaboStar^TM^ 4-DI/UV water system (LabostarTM TWF, Evoqua Water Technologies LLC, Warrendale, PA, USA). Potassium chlorate (KClO_3_ > 99.00%) and nitric acid (HNO_3_ ≥ 99.00%) from Sigma-Aldrich (MO, USA) and 13 mM sodium dodecyl sulfate (SDS > 99.0%) from Fluka (Wayne, PA, USA) were used. In addition, 0.7 wt % of natural rubber (NR) latex in ammonia solution was purchased from Chilean rubber and a latex ESPO products supplier. PVA (>99.00%) from Sigma-Aldrich (MO, USA) was used at a concentration of 11% *w/v* and AG from an acacia tree from Sigma-Aldrich (MO, USA) was used at a concentration of 10% *w*/*v*. Two concentrations of TrGO, 0.5% *w*/*v* and 1% *w*/*v*, with respect to the mass of the different components of the ES solution were used.

The instruments used in this study were as follows: The Raman spectra were obtained on a Renishaw PLC, Wotton-under-Edge UK equipment with a laser wavelength of 514.5 nm. X-ray photoelectron spectroscopy (XPS) was carried out using a Perkin Elmer XPS-Auger spectrometer, model PHI 1257 (Waltham, MA, USA). This equipment includes an ultra-high vacuum chamber, a hemispheric electron energy analyzer, and an X-ray source with Kα radiation unfiltered from an Al (hν = 1486.6 eV) anode. The SEM-EDS analysis of the EM was performed in a JEOL JSM-IT300LV microscope (JEOL USA Inc., Peabody, MA, USA) that was connected to an energy-dispersive X-ray detector for elemental analysis with computer-controlled Aztec EDX system software from Oxford Instruments, Abingdon, UK. XRD analysis was obtained on a Siemens D-5000 powder X-ray diffractometer with a CuKα radiation (1.54 Å).

### 2.2. Synthesis of Graphene Oxide and Thermally Reduced Graphene Oxide

The synthesis of graphene oxide (GO) was carried out by adding 5 g of graphite to 100 mL of fuming HNO_3_ in a refrigerated reactor at 0 °C, and the mixture was left under stirring. Then, 40 g of KClO_3_ were slowly added to the mixture, and it was left to react for 22 h. Then, the mixture was poured into 1500 mL of distilled water. The solid product was washed several times with distilled water until it reached pH 7.0 and was separated by centrifugation and dried at 70 °C for 12 h. The resulting GO was thermally reduced in a MTI 1200X tube furnace under continuous flow of argon (200–500 mL/min) to obtain thermally reduced graphene oxide (TrGO). For this, an initial heating ramp of 20 °C/min was considered until reaching 200 °C. Then, a heating ramp of 10 °C/min was used between 200 and 1000 °C, kept at 1000 °C for 2 min, and then allowed to cool to room temperature.

### 2.3. Preparation of Electrospun Fiber Meshes by Electrospinning

Electrospun meshes (EM) were prepared in the presence and absence of TrGO (control). TrGO was loaded in the EM at concentrations of 0.5% or 1%, respectively. The choice of the two TrGO concentrations was that it is not possible to prepare suitable polymeric solutions for an ES process containing more that 1% TrGO. Concentration greater that 1% TrGO results in a poor dispersion of TrGO in the polymer fibers composite. Additionally, a higher content of TrGO require the use of high frequency and intensity ultrasound for long periods. It is known that high-power ultrasound induces a temperature elevation of the polymer solution resulting in the degradation of polymeric materials and preventing a proper ES process. Different EM (EM1–EM4) were prepared as follows: EM1 (PVA), EM2 (PVA + NR + SDS), EM3 (PVA + NR + SDS + AG), and EM4 (PVA + NR + SDS + AG + TrGO). Polymer composite fibers with different fiber orientations, random and aligned fiber topologies, were prepared using flat plate and rotating drum collectors, respectively. All EM were obtained by using the following ES parameters: the flow rate and electric potentials used for the preparation of EM were in the range of 15–19 kV and 1000–1400 µL/h, respectively. The distance of the collecting plate covered by aluminum foil from the metallic needle was varied from 15 to 18 cm. In the case of EM with aligned fibers, the rotation speed of the rotating drum was 1700 rpm (clockwise).

### 2.4. Hyperthermia Evaluation

The hyperthermia property of EM was evaluated varying the distance from 2 to 6 cm and electric potential from 5 to 10 V at a constant time of 30 s under 808 nm laser irradiation with a LRD-0808 Laserglow^®^ equipment using a thermal imager (Fluke^®^ TiS20, Laserglow COM Limited, Toronto, ON, Canada). All photothermal images were analyzed using a SmartView^®^ software.

## 3. Results and Discussion

### 3.1. Characterization of Thermally Reduced Graphene Oxide

#### 3.1.1. Raman Spectroscopy

Figure 1 shows the Raman spectra of graphite and TrGO samples. Spectrum (a) of Figure 1 corresponds to the Raman spectrum of graphite in which the graphitic band (G band), corresponding to the first order vibration mode of E_2g_ phonon of the sp^2^ carbon atoms, appears at 1590 cm^−1^. Additionally, a very small D band is observed at 1350 cm^−1^, which is typically associated to the breathing modes of aromatic carbon rings. This band is attributed to the edge defect and functional groups present in the lattice of graphene materials [28,29]. The 2D band, which is the overtone of D bands is seen ca. 2700 cm^−1^, while the overtone of D and D’ bands, the so-called D+D’ band, appears at 2920 cm^−1^ [30]. Spectrum (b) of Figure 1 shows a small D band at 1350 cm^−1^ and the G band at 1572 cm^−1^ of TrGO. A drastic intensity decrease of the 2D band at 2710 cm^−1^ was observed. This could be attributed to the order loss and increase of defects, which inhibited the overtone occurrence. Another important change is related to the drastic increase of the D band, which usually is associated to the increase of edge defects vacancies and distortions as well as the functionalities in graphene lattice. The ratio of the intensity of the G to D bands (I_D_/I_G_) gives an indication of the degree of defects in the graphene structure. The intensity ratio of D to G bands (I_D_/I_G_) for graphite is 0.03 (Figure 1a), while for TrGO, it is 0.89 (Figure 1b). This fact indicates the drastic increase of the defects of the graphene lattice as a consequence of the consecutive oxidation and reduction processes in which the graphite is subjected in order to obtain TrGO [31].

On the other hand, the broadening of the G band of TrGO as compared with the G band of the graphite indicates the random interactions of the TrGO sheets. Several researchers have reported that the thermal process facilitates the elimination of oxygenated functional groups and concomitantly favors the restoration of the long-range conjugated-π system. This fact is associated with the increase of thermal and electrical conductivity. Likewise, this long-range conjugated-π system imparts to graphene materials a stable absorption rate and absorption bandwidth in a large incident angle range in near infrared light [32]. Consequently, the use of TrGO as a filler in polymer fibers for hyperthermia applications could be considered as a valuable therapy option.

This section may be divided by subheadings. It should provide a concise and precise description of the experimental results, their interpretation, as well as the experimental conclusions that can be drawn.

#### 3.1.2. X-ray Diffraction

Figure 2 shows the XRD patterns of graphite (a) and TrGO (b) samples. A narrow and intense diffraction peak ca. 2θ = 26.3° associated with the graphitic (002) plane is seen for graphite. By using the Bragg’s law [33], the interlayer distance can be estimated, which typically is associated with the distance between graphene layers. In the case of graphite, this interlayer distance corresponds to 3.34 Å, which is consistent with values reported in the literature [29]. Conversely, the diffraction peak of TrGO is significantly wider than that of graphite and appears at 2θ = 25.4°, corresponding to an interlayer distance of 3.51 Å.

On the other hand, the crystallite size and the number of stacked layer of graphene in these materials can be determined by using the Scherrer equation [34]. In this regard, the crystallite size and the number of stacked graphene layers of TrGO were 13.6 Å and 5, respectively [35]. This is significantly lower than the values for graphite, which present a crystallite size and the number of stacked layers of 180 Å and 54, respectively.

As mentioned, the (002) peak of TrGO is significantly wider than that observed for graphite. This is attributable to the partial loss of crystallinity, which is a result of the oxidation and reduction processes, where the latter causes the exfoliation of the graphene layers [36,37]. It is also worth noting that according to Bianco et al. [35], a graphene material is considered to be a material that contains up to 10 stacked layers. TrGO, which is composed of only five stacked layers, can be considered as a few-layered reduced GO. However, the interplanar distance of TrGO (3.51 Å) is greater than that of its precursor, graphite (3.34 Å). These results are due to a partial remanence of functional groups in TrGO.

#### 3.1.3. X-ray Photoelectron Spectroscopy

Figure 3 presents the photoelectric lines C 1s (Figure 3a) and O 1s (Figure 3b) of TrGO. The functional groups contributions were estimated by using curve fitting, which based on a Shirley baseline and Lorentz model. The assignments of these functional groups are presented in Table 1. The O/C ratio evidenced the low content of oxygen atoms in TrGO, indicating that the thermal process favors the elimination of oxygenated functional groups. The contribution observed at 286.2 eV corresponds to carbonyl groups (Figure 1a), while that observed at 287.5 eV corresponds to the contribution of alcohol and the ether functional groups present in the graphene structure. The latter is the dominant contribution of the oxygenated functional groups. Similarly, in an O 1s photoelectric line, the dominant contribution observed at 533.7 eV (Figure 1b) corresponds to alcohol and ether functional groups. The lower contribution observed at 531.3 eV is associated with ketone groups. Although the TrGO possesses a higher content of C=C and C-C bonds that have a nonpolar character, the presence of oxygenated functional groups such as ether, alcohol, or ketone, suggests a partial affinity of TrGO with polar polymers such as PVA.

### 3.2. Characterization of Electrospun Meshes

#### 3.2.1. Scanning Electron Microscopy

Figure 4 shows optical (a,b) and scanning electron microscopy (c-f) images of random (c) and aligned (d,e) EM4 and aligned EM3 (f). Figure 4a,b shows the optical images of general fiber orientations of EM4, and Figure 4c,d shows SEM images of random and aligned composite EM4 fibers showing the presence of beads. Meanwhile, Figure 4e,f shows EM4 and EM3 with aligned fibers obtained in the presence and absence of TrGO as a NIR-active thermal inducer. Figure 4e shows EM4 fibers composed of PVA, NR, and GA polymers which presented fibers with more homogeneous surface roughness than EM3 and smaller fiber diameters ranging from ≈80 nm to ≈300 nm. The presence of TrGO in this EM can be identified as dark areas that can be transversely and longitudinally distinguished along the composite aligned fibers and highlighted by the white circles in Figure 4e. At the same time, Figure 4f shows EM3 fibers composed of PVA, NR, and GA but without TrGO, as demonstrated by the absence of TrGO in the composite fibers.

Moreover, Figure 5 shows the morphology of EM2 with aligned fibers containing PVA, NR, and SDS utilized as surfactant as in EM3 and EM4, in the absence (a,c) and in the presence of TrGO (b,d). Figure 5a,c shows polymer composite fibers with rougher surface and wider diameters ranging from ≈1 to 2 µm. In general, fibers of EM2 without beads were found. However, when the EM2 fibers were prepared in the presence of TrGO (Figure 5b,d), fibers with smoother surface and narrower diameters ranging from 180 to 300 nm were obtained. In Figure 5d, the presence of TrGO can be identified as dark areas along this composite polymer fiber, as was also shown by the white circles in Figure 4e for EM4.

#### 3.2.2. Elemental Analysis Using Scanning Electron Microscopy

Figure 6 shows the SEM-EDS analysis of EM4. We found a coherent elemental composition with respect to the composition of EM4 polymer composite. Thus, a high content of the carbon (66.6%) followed by aluminum (Al), oxygen (5.9%), and sulfur (0.2%) was identified. The presence of Al is due to the use of the aluminum foil to cover the surface of the rotating collector, where the fibers were deposited. The presence of sulfur is due to the use of SDS in the preparation of the fibers.

#### 3.2.3. X-Ray Diffraction Analysis of Fibers

Figure 7 shows the X-ray diffraction analysis of the EM3 and EM4 samples. The strong peak observed for both EM at 2θ = 17.1° corresponds to the PVA diffraction plane due to its semi-crystalline nature. Moreover, the wide peak observed at 2θ = 23.0° could be due to the partial loss of crystallinity of PVA as well as the presence of TrGO in these fibers. The partial loss of the crystallinity could be the result of the interaction of PVA and AG through hydrogen bonding [38]. The presence of oxygenated functional groups in TrGO promotes its interaction with PVA and AG through hydrogen bonding favoring its permanence in the electrospun fibers.

#### 3.2.4. Hyperthermia Evaluation of the Electrospun Meshes

The hyperthermia evaluation was performed on EM3 and EM4 samples in the absence and presence of TrGO respectively, in order to determine the temperature they reach when irradiated with an NIR laser. Moreover, during the hyperthermia evaluation of composite fiber, the distance (2, 4, and 6 cm) and electric potential (5, 7.5, and 10 V) parameters were considered. The thermal evaluation carried out on EM3 is shown in Table 2 and those for EM4 with 0.5% TrGO and EM4 with 1% TrGO are shown in Table 3 and Table 4, respectively. Table 2 shows that the registered thermal values recorded for EM3 were on average 19 °C, 19.4 °C, and 20.4 °C for random and 18.8 °C, 19.3 °C, and 20.1 °C for aligned composite fibers at electric potentials of 5, 7.5, and 10 V, respectively. These experimental results demonstrate that for both fiber orientations, there was no effect of electric potential or distance when the EM does not contain TrGO as an NIR-active thermal inducer.

On the contrary, in both EM4 with random and aligned fibers containing 0.5% TrGO (Table 3), the infrared thermography shows that the thermal radiation capacity increases, reaching average thermal values from 18.3 to 37.8 °C and from 18.4 to 33 °C, respectively. We found that the recorded thermal values of random EM4 in average were 19.6, 28.9, and 33 °C for electric potentials of 5, 7.5, and 10 V, respectively, demonstrating the direct proportional effect of electric potential and that the hyperthermia was inversely proportional to distance when using 0.% TrGO as an NIR-light induced photothermal agent. In the case of EM4 with aligned fibers, we observed that the registered thermal values were on average 19.5, 26.4, and 29.2 °C for the electric potentials of 5, 7.5, and 10 V, respectively, showing that there was an increasing tendency of average temperature with the electric potential and that its values are lower compared to EM4 with random fibers orientation. Our experimental findings suggest that a random fiber orientation results in greater hyperthermia, which in turn is inversely proportional to distance.

The infrared thermography of both EM4 with random and aligned fibers containing 1% TrGO are shown in Table 4 demonstrating that EM loaded with a higher concentration of TrGO further increases the thermal infrared capacity, finding average values from 18.4 to 52.3 °C for random fibers and from 18.1 to 49.3 °C for aligned fibers, respectively. Table 4 shows that the thermal infrared values are on average 19.8, 32.3, and 43.4 °C for electric potentials of 5, 7.5, and 10 V, respectively. Again, we found a direct dependence of average temperature with electric potential values and that the hyperthermia capacity is inversely proportional to the distance when 1% TrGO was used. We also found that the registered thermal values were on average 19.2, 31.8, and 41.6 °C for the electric potentials of 5, 7.5, and 10 V, respectively. The results presented in Table 3 suggest that EM with random fiber orientation displays a greater hyperthermia capacity than polymer composite with an aligned fibers orientation.

Figure 8a–d shows the thermographic images obtained from EM3 and EM4 with random and aligned fibers containing 0.5% and 1% TrGO registered at 10 V and 2 cm after NIR irradiation. Figure 8a,b shows that regardless of the irradiation conditions tested, the hyperthermia of the EM3 did not increase, always showing values of ≈21 °C. However, in the case of EM4 using two concentrations of TrGO, the recorded temperature values were much higher (Figure 8c–f), these being higher with a higher concentration of TrGO and higher with increasing the applied electric potential.

On the other hand, for the EM4 of random and aligned fibers with 1% TrGO and at a high electric potential of 10 V, it was observed that there was a compromise regarding the distance to reach values of 52.3 °C at 2 cm and 48.6 °C at 2 cm, respectively, which are adequate and scarcely superior to what is required for an optimal 47 °C in vivo photothermal treatment [5].

According to the standard deviations (SD) for EM4 presented in Table 3 (4.33 and 2.26) and Table 4 (3.96 and 2.90), it is possible to establish the topology with less temperature variation between selected irradiation points. These variations were analyzed and confirmed through the statistical measure of SD. These values indicate that for EM4 loaded with 0.5% and 1% TrGO, the thermal variations are smaller in aligned fibers with respect to the random fibers.

On the other hand, Table 5 shows the averages of the thermal variation (ΔT) of the infrared thermography registered for EM3 and EM4 with random fibers orientation. We found higher thermal values for EM4, and the ΔT suggests that the concentration of 1% TrGO in EM4 is essential when irradiating with NIR to promote hyperthermia.

Additionally, Figure 9 shows the behavior of the ΔT obtained between the EM3 and EM4 considering only random fibers orientation with respect to the applied electric potential and distance. Our results showed that there was a decrease in ΔT as the irradiation was carried out at a greater distance with higher electric potential, which implies that the greater ΔT difference was due to the increase in the conversion of optical to thermal energy by TrGO in the EM4 at the highest electric potential. Furthermore, at different distances, a decrease in ΔT was observed, as the applied electric potential was lower.

In order to analyze the effect of fiber orientations of EM on thermal irradiation capacity, the infrared thermography for the EM4 was carried out using the higher concentration of TrGO for both topologies, that is, random and aligned fibers orientation, as is shown in the Table 6. We found that the EM4 with random fibers orientation had higher thermal values compared to aligned fibers orientation, and in general, there were lower ΔT values at smaller distances. These experimental findings show the key role of the fibers orientation in the EM as well as the electric potential and distance parameters to achieve a representative and homogeneous thermal property demonstrating the viable heat capacity and thermal conductivity of EM for hyperthermia application.

In this sense, Figure 10 shows the ΔT behavior recorded for EM4 with 1% TrGO with random and aligned fibers orientation varying the electric potential and distance parameters. In general, a decreasing behavior of the ΔT values was observed, which was limited by the different applied distances, where the proximity of the NIR laser to the EM surface favors the hyperthermia properties of TrGO, favoring the increase in ΔT as a function of electric potential, as it is reflected in distances of 2 and 6 cm, respectively. Surprisingly, we found that the ΔT behavior changed at 4 cm, so a more homogeneous character is suggested by both EM topologies, balancing the ΔT when irradiating at different applied electric potentials. In turn, at highest electric potential of 10 V, the behavior of ΔT was greater at the extremes of the distances, having a less value of 1.3 at a distance of 4 cm.

## 4. Conclusions

Electrospun polymer composites were prepared using natural and synthetic polymers with random and aligned fibers orientations loaded with 0.5% or 1% of TrGO and their hyperthermia properties by NIR thermography changing the distances from 2 to 6 cm and electric potential from 5 to 10 V were evaluated. Raman, XPS, XRD, and SEM-EDS confirmed the preparation of TrGO, the composition of the polymer fibers composite, and the presence of TrGO along the fiber in the EMs. The number of graphene layers as well as the interplanar distance before and after the thermal reduction processes were determined through the Bragg and Scherrer equations. We found that SDS used as surfactant facilitated the formation of fibers in EM due to its surface-active property, and it also provided greater stability to the colloidal suspension of the NR latex, reducing the surface tension and facilitating the ES process. Moreover, arabic gum improved the thermal resistance of the resulting EM due to its crosslinking properties. The infrared thermography on the EM4 revealed that TrGO is an effective photothermal agent achieving a suitable EM material for its possible use in photothermal therapy. The fibers orientation, the distance of laser irradiation, and the applied electric potential during infrared thermography were crucial parameters to determine the properties of infrared thermography of polymer composite fibers. It was also confirmed that the thermography was smaller in EM composed of aligned fibers that random fibers using 0.5% and 1.0% TrGO.

## Figures and Tables

**Figure 1 polymers-12-02663-f001:**
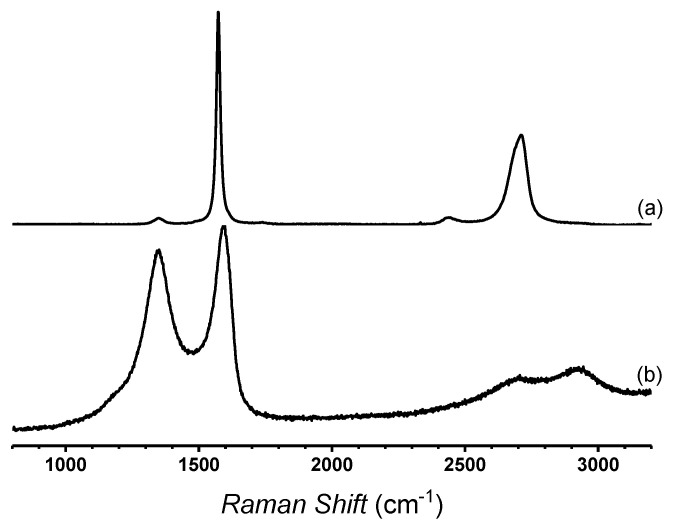
Raman spectra of graphite (**a**) and thermally reduced graphene oxide (TrGO) (**b**).

**Figure 2 polymers-12-02663-f002:**
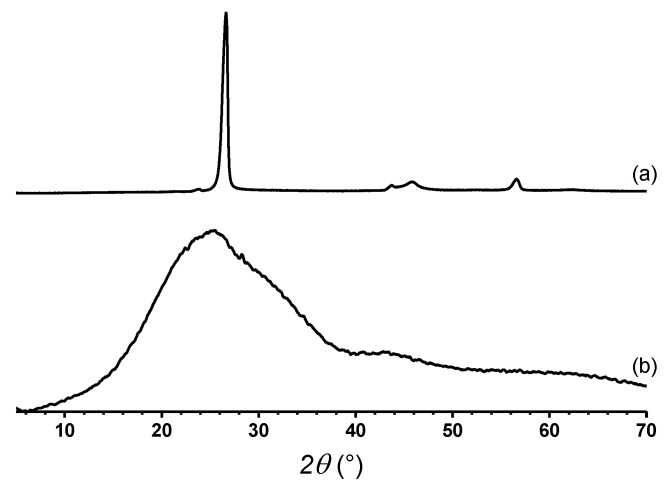
XRD patterns of graphite (**a**) and TrGO (**b**).

**Figure 3 polymers-12-02663-f003:**
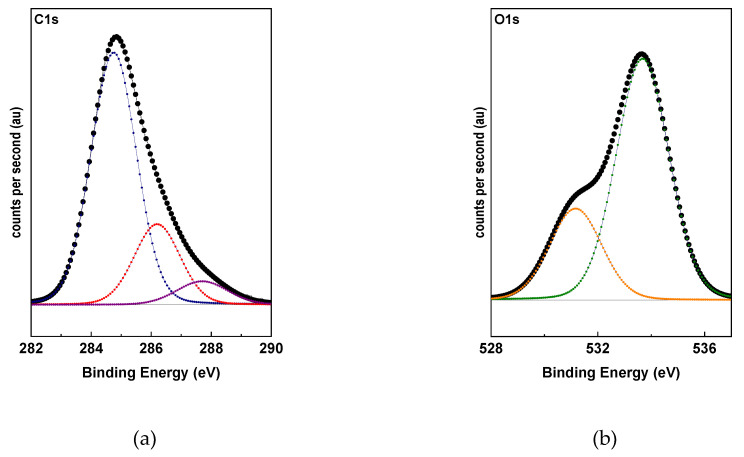
XPS of C 1s (**a**) and O 1s (**b**) photoelectric lines of TrGO.

**Figure 4 polymers-12-02663-f004:**
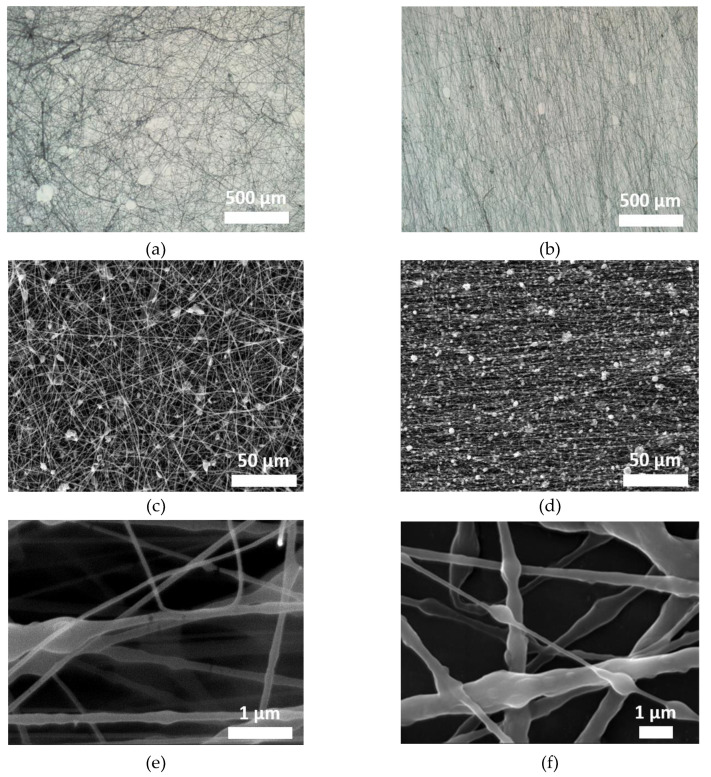
Optical microscopy (**a**,**b**) and SEM images of random (**c**) and aligned (**d**) EM4. The white circles indicate the presence of TrGO in the aligned fibers of EM4 (**e**) and the absence of TrGO in aligned fibers in EM3 (**f**).

**Figure 5 polymers-12-02663-f005:**
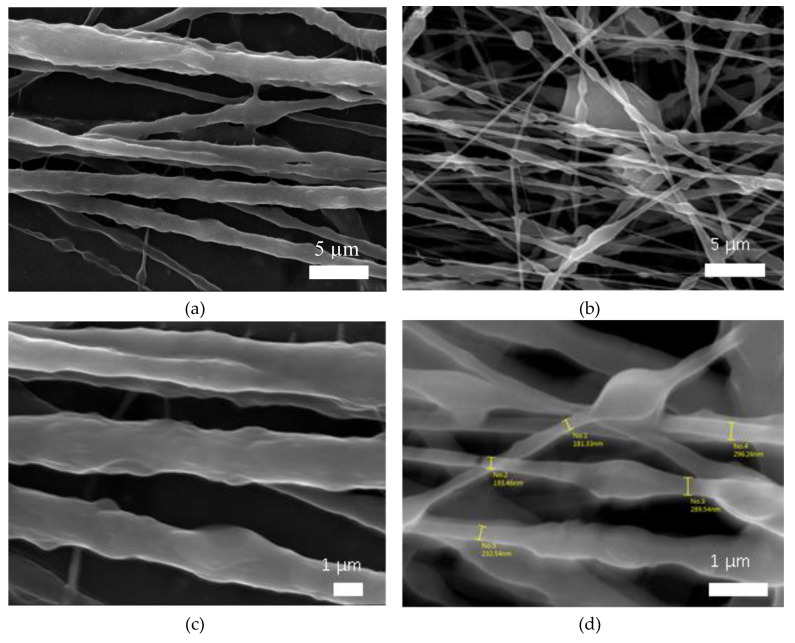
SEM images of aligned fibers of EM2 in the presence (**a**,**c**) and absence (**b**,**d**) of TrGO. The white circles indicate the presence of TrGO in the aligned fibers of EM2 (**d**). EM: electrospun mesh.

**Figure 6 polymers-12-02663-f006:**
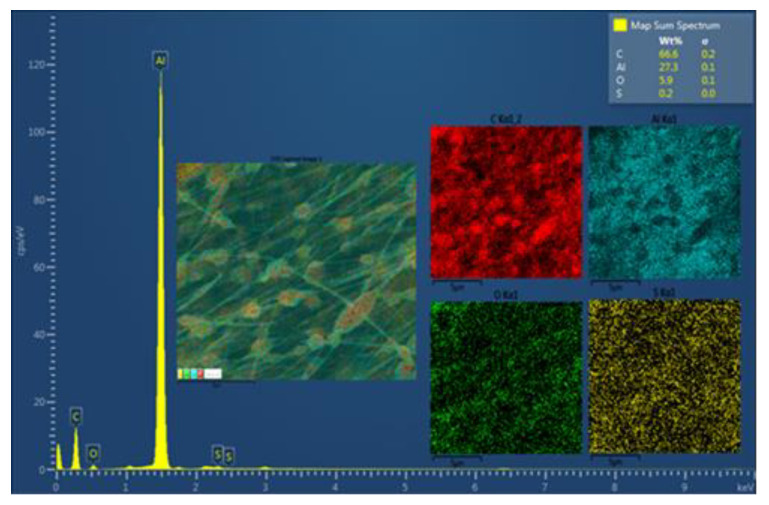
SEM-EDS of aligned EM4. The elements of carbon (C), aluminum (Al), oxygen (O), and sulfur (S) are designated with the colors red, light blue, green, and yellow, respectively.

**Figure 7 polymers-12-02663-f007:**
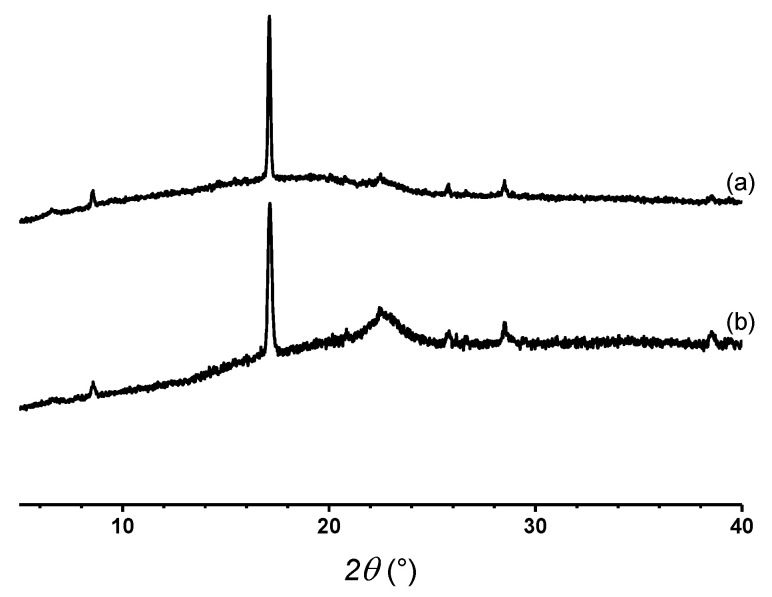
XRD patterns of EM3 (**a**) and EM4 (**b**).

**Figure 8 polymers-12-02663-f008:**
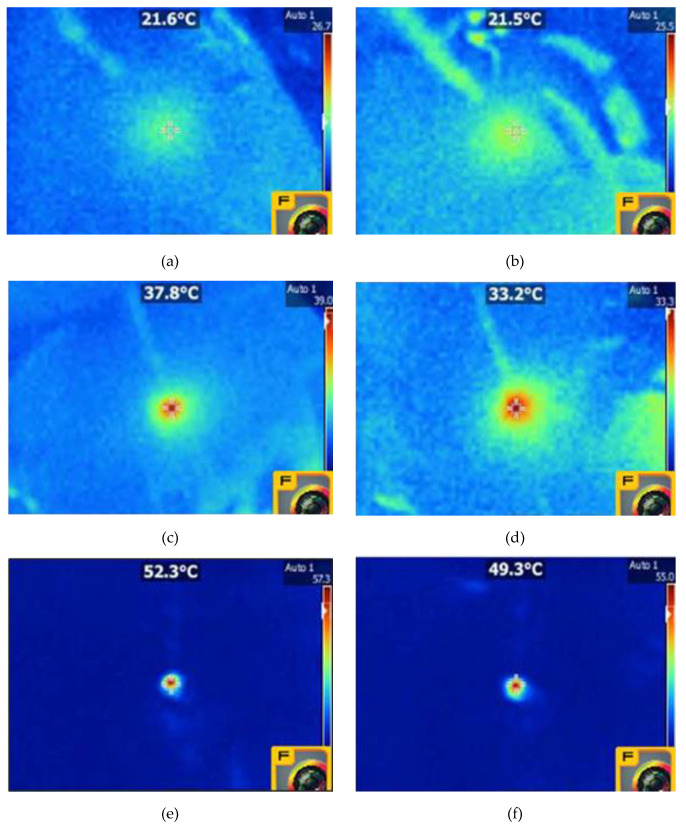
Infrared thermography of EM3 with random and aligned fibers and EM4 with 0.5% and 1% TrGO registered at 10 V and 2 cm: (**a**) EM3 with random fibers, (**b**) EM3 with aligned fibers, (**c**) EM4 with random fibers (0.5% TrGO), (**d**) EM4 with aligned fibers (0.5% TrGO), (**e**) EM4 with random fibers orientation (1% TrGO), and (**f**) EM4 with aligned fibers (1% TrGO).

**Figure 9 polymers-12-02663-f009:**
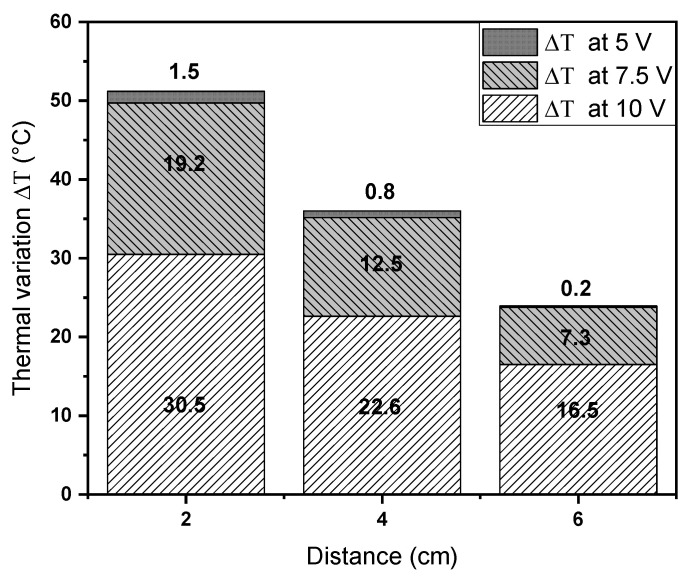
Thermal variation (ΔT) of the average temperature obtained in random fibers of EM3 and EM4 varying the irradiation distance and electric potential.

**Figure 10 polymers-12-02663-f010:**
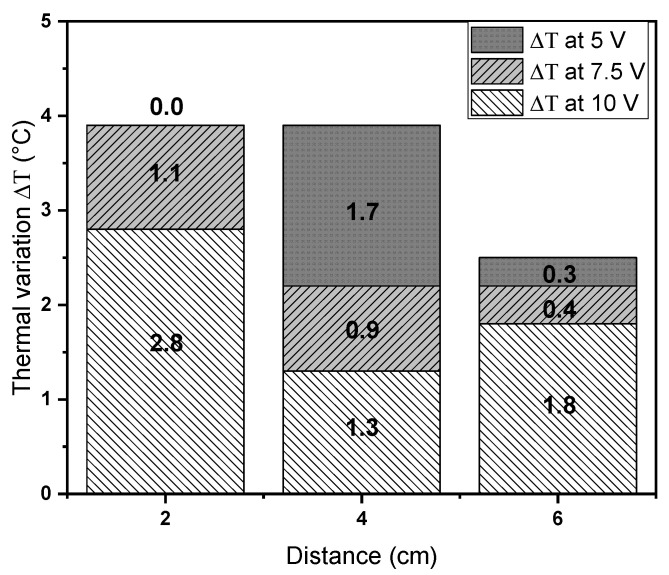
Thermal variation (ΔT) of the average temperature obtained in random and aligned fibers of EM4 varying the irradiation distance and electric potential.

**Table 1 polymers-12-02663-t001:** O/C ratio, binding energy, assignment, and area peak of photoelectric lines in TrGO.

Sample	Signal	Binding Energy (eV)	Assignments	Area (%)	O/C Ratio
TrGO	C 1s	284.8	C = C sp^2^ C-C sp^3^	70	0.053
		286.2	C*-*O*-C*/C*-*O*-H	23	
		287.5	-C* = *O*	7	
	O 1s	531.3	-C = *O**	27	
		533.7	C-*O**-C/C-*O**-H	73	

**Table 2 polymers-12-02663-t002:** Infrared thermography of EM3 with random and aligned fibers recorded at different electric potentials and distances.

Electric Potential (V)	Distance (cm)	Average Temperature (°C) ^a^	Standard Deviation ^a^	Average Temperature (°C) ^b^	Standard Deviation ^b^
5	2	19.2	0.57	19	0.55
	4	19.2	0.31	19	0.26
	6	18.6	0.25	18.5	0.32
7.5	2	19.8	0.25	19.8	0.38
	4	19.4	0.21	19.2	0.21
	6	18.9	0.44	18.9	0.44
10	2	21.2	0.4	20.9	0.57
	4	20.3	0.12	19.9	0.32
	6	19.7	0.45	19.4	0.21

The average temperature was obtained from three temperature measurements. Superscripts a and b indicate EM3 with random and aligned fibers, respectively.

**Table 3 polymers-12-02663-t003:** Infrared thermography of EM4 with random and aligned fibers containing 0.5% TrGO recorded at different electric potentials and distances.

Electric Potential (V)	Distance (cm)	Average Temperature (°C) ^a^	Standard Deviation ^a^	Average Temperature (°C) ^b^	Standard Deviation ^b^
5	2	20.7	0.36	20.5	0.26
	4	19.5	0.46	19.3	0.21
	6	18.6	0.31	18.7	0.35
7.5	2	34.3	0.46	30.7	0.3
	4	29.2	0.56	26	0.21
	6	23.3	0.26	22.6	0.15
10	2	37.3	0.47	33.1	0.1
	4	33	0.7	29.5	0.35
	6	28.7	0.75	25	0.32

The average temperature was obtained from three temperature measurements. Superscripts a and b indicate EM4 with random and aligned fibers, respectively.

**Table 4 polymers-12-02663-t004:** Infrared thermography of EM4 with random and aligned fibers containing 1% TrGO recorded at different electric potentials and distances.

Electric Potential (V)	Distance (cm)	Average Temperature (°C) ^a^	Standard Deviation ^a^	Average Temperature (°C) ^b^	Standard Deviation ^b^
5	2	20.7	0.65	20.7	0.4
	4	19.9	0.45	18.4	0.26
	6	18.9	0.31	18.4	0.25
7.5	2	39	0.15	37.9	0.31
	4	31.7	0.45	31.8	0.31
	6	26.3	0.5	25.8	0.15
10	2	51.3	0.71	48.6	0.32
	4	42.6	0.42	41.7	0.45
	6	36.2	0.32	34.4	0.45

The average temperature was obtained from three temperature measurements. Superscripts a and b indicate EM4 with random and aligned fibers, respectively.

**Table 5 polymers-12-02663-t005:** Thermal variation (ΔT) between the infrared thermography values recorded for EM4 and EM3 with random fibers orientation.

Electric Potential (V)	Distance (cm)	Average Temperature (°C) ^a^	Average Temperature (°C) ^b^	ΔT (°C)
5	2	20.7	19.2	1.5
	4	20.1	19.2	0.9
	6	18.7	18.6	0.1
7.5	2	39.0	19.8	19.2
	4	32.0	19.4	12.6
	6	26.2	18.9	7.3
10	2	51.7	21.2	30.5
	4	43.0	20.3	22.7
	6	36.2	19.7	16.5

The average temperature was obtained from three temperature measurements. Superscripts a and b indicate EM4 and EM3 with random fibers orientation, respectively.

**Table 6 polymers-12-02663-t006:** Thermal variation (ΔT) between the infrared thermography values recorded for EM4 with random and aligned fibers orientation.

Electric Potential (V)	Distance (cm)	Average Temperature (°C) ^a^	Average Temperature (°C) ^b^	ΔT (°C)
5	2	20.7	20.7	0
	4	20.1	18.4	1.7
	6	18.7	18.4	0.3
7.5	2	39.0	37.9	1.1
	4	32.0	31.1	0.9
	6	26.2	25.8	0.4
10	2	51.7	48.9	2.8
	4	43.0	41.7	1.3
	6	36.2	34.4	1.8

The average temperature was obtained from three temperature measurements. Superscripts a and b indicate EM4 with random and aligned fibers orientation, respectively.

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
