# Peer review of "In Vitro Hyperthermia Evaluation of Electrospun Polymer Composite Fibers Loaded with Reduced Graphene Oxide"

_polymers, 2020, doi:10.3390/polym12112663_

Round 1

Reviewer 1 Report

The manuscript by Zárate et al. reports on the evaluation of the hyperthermia properties of electrospun polymer composite fibers loaded with TrGO. The topic of the manuscript is worthy of investigation, well fits with teh scope a juornal like polymers and is of interest for scientists working in either the macromolecular or materials science chemistry fields, nevertheless it is an opinion of this reviewer that a revision is needed before further evaluation of the study. In details, authors should better explain: a. the novelty of their study and the expected and obtained advantages of the proposed system compared to similar materials available in the literature. b. The rationale for choosing TrGO as functional element among the available graphene derivatives c. 0.5 and 1.0% as TrGO concentrations to be loaded in the EM.

Reviewer 2 Report

In vitro hyperthermia evaluation of electrospun polymer composite
fibers loaded with reduced graphene oxide is a nicely written paper. I must appriciate the flow of content one can observe while reading the paper. All the results are very adequately described without leaving any confusion to the readers. It can be accepted after minor revisions and following points can help improve it further.

In the last paragraph of introduction please add few lines about the importance of your work. Moreover, generally in the introduction very less background literature is provided. It will be nice if you cite some very recent papers on the relevant subject so one can understand well the importance of your work and which gaps have been filled.

There are a number of typos that need to be written in subscript or superscript. for example Eg2 and Sp2 , cm-1etc.

Reference 5,6 can be replaced with other reference as it is self citation and does not belongs very importantly to be mentioned.Same is the case for ref. 15-16.

Round 2

Reviewer 1 Report

Authors well addressed the reviewer comments and improved the manuscript accordingly. This reviewer is recommending publication in its current form.